

# Gastric *Helicobacter pylori* infection perturbs human oral microbiota

Eng-Guan Chua[1,*], Ju-Yee Chong[1,*], Binit Lamichhane[1], K. Mary Webberley[1], Barry J. Marshall[1,2], Michael J. Wise[1,3] and Chin-Yen Tay[1,2]

[1] The Marshall Centre for Infectious Diseases Research and Training, University of Western Australia, Crawley, Western Australia, Australia
[2] Shenzhen Dapeng New District Kuichong People Hospital, Shenzhen City, Guangdong Province, China
[3] Computer Science and Software Engineering, University of Western Australia, Crawley, Western Australia, Australia
[*] These authors contributed equally to this work.

## ABSTRACT

**Background**. We investigated the effects of gastric *Helicobacter pylori* infection on the daytime and overnight human oral microbiota.

**Methods**. Twenty four volunteers were recruited. Ten tested positive for *H. pylori* infection by the Carbon-14 Urea Breath Test, and the rest were negative. Two oral swabs were collected: one immediately after waking up in the morning and before brushing teeth, and another in the evening before teeth-brushing. DNA extract acquired from each swab was subjected to Illumina sequencing of *16S rRNA* gene amplicons. The microbial abundance and composition were analysed in relation to *H. pylori* infection status.

**Results**. *Helicobacter pylori*-positive individuals had significant changes in the alpha and beta diversities in the daytime samples in comparison to those who were *H. pylori* negative. To identify which taxa could be significantly affected within the cohorts in the daytime, we employed the LEfSe method. When compared against UBT-negative samples, significantly higher abundances were detected in both *Pseudomonas* and *Roseomonas*, while *Fusobacterium*, *Solobacterium*, *Haemophilus* and *Streptococcus* were significantly decreased in the UBT-positive samples.

**Discussion**. Our data demonstrated that *H. pylori* infection affects the human daytime oral microbiota. The hitherto undocumented changes of several bacterial genera due to *H. pylori* infection require more studies to examine their potential health effects on affected individuals.

Corresponding author
Eng-Guan Chua,
eng.chua@uwa.edu.au

## INTRODUCTION

*Helicobacter pylori* is a highly adapted gastric pathogen that infects nearly half of the world's population. It is an important etiological factor for the development of peptic and duodenal ulcerations, chronic gastritis and gastric atrophy. In more severe, but less common cases, *H. pylori* infection causes gastric adenocarcinoma and gastric mucosa-associated lymphoid tissue (MALT) lymphoma (*Marshall & Windsor, 2005*).

Although *H. pylori* is a well-known human gastric pathogen, it is less commonly regarded as a constituent of the oral cavity flora. However, it is listed in the Human Oral Microbiome Database (*Chen et al., 2010*). Multiple previous PCR studies have detected *H. pylori* based on *16S rRNA*, *cagA* and *glmM* gene sequences in the saliva and dental plaque samples (*Amiri et al., 2015*; *Castro-Munoz et al., 2017*; *Silva et al., 2009*). More importantly, several groups have reported the presence of *H. pylori* in the oral cavity by culture, indicating that the human oral cavity can act as a reservoir for *H. pylori* and therefore a potential source for transmission (*Agarwal & Jithendra, 2012*; *Goosen et al., 2002*; *Teoman et al., 2007*; *Wang et al., 2014*).

Following the lead of the earlier studies, we set out to detect *H. pylori* in the oral cavity. We used 16S metagenomic sequencing to examine the presence of this bacterium in the oral cavities of individuals who had their gastric infection status ascertained by the Urea Breath Test. We hypothesised that the prevalence or abundance of *H. pylori* may vary with the time of day and so took samples at two time points: in the morning before tooth brushing and in the evening. It is thought that *H. pylori* might be predominantly deposited in the mouth by reflux overnight while hosts were lying down, but that it may be removed by tooth brushing. We further hypothesised that gastric and/or oral *H. pylori* infection may lead to wider changes in the human oral microbiome.

By testing at two times of day, we were also able to study the effect of *H. pylori* on what we thought might be two different microbial groups. We expected the early morning sampling to reveal the relationship between *H. pylori* infection and the predominantly anaerobic or facultative anaerobic bacteria present in the oral cavity. We believed that during night time, when one is asleep and is breathing normally via the nasal route, the oral cavity would remain relatively closed for a prolonged period, generating a microaerobic or anaerobic condition that favours the growth of anaerobes or facultative anaerobes (*Fitzpatrick et al., 2003*). The other samples were taken in the evening when the oral cavity is anticipated to be in an aerobic state due to the mouth opening for activities such as eating and drinking, and talking. This design allowed us to investigate the potential effect of *H. pylori* infection to influence the structure of both the anaerobic and aerobic bacterial communities. These results would enable us to further assess the changes, if any, in oral microbiome on the health of human host.

## MATERIAL AND METHODS

### Ethics approval and consent to participate

This study was approved by the Human Ethics of the University of Western Australia (RA/4/1/7953). Written consent was obtained from all individuals participated in this study.

### Sample collection

A total of 24 University of Western Australia postgraduate students who brushed their teeth twice a day (after waking up and before going to bed) were recruited over a 6-month period and screened for *H. pylori* infection using the PYtest Urea Breath Test (UBT) kit (Tri-Med, Western Australia, Australia). Two oral swabs, rotated along the inside of the cheek for

approximately 10 s, were collected by study participants: one in the morning immediately after waking up and another at six in the evening, both before teeth-brushing. The morning samples allowed us to characterise the oral microbial community composition of the post overnight cycle whilst the latter focused on the daytime oral microbiome. For consistency, the morning and evening samples are referred to as the overnight and daytime samples, respectively.

## 16S amplicons generation and sequencing

Oral swab DNA extraction was performed using DNeasy® blood and tissue kit (Qiagen, Hilden, Germany). The V3-V4 region of the 16S rRNA gene was amplified from each DNA extract using the S-D-Bact-0341-b-S-17 (5′-CCTACGGGNGGCWGCAG-3′) and S-D-Bact-0785-a-A-21 (5′-GACTACHVGGGTATCTAATCC-3′) primer pair (*Klindworth et al., 2013*). Illumina adapter overhang sequences, 5′-TCGTCGGCAGCGTCAGATGTGT ATAAGAGACAG-3′ and 5′-GTCTCGTGGGCTCGGAGATGTGTATAAGAGACAG-3′, were added to the 5′ends of the forward and reverse primers, respectively. Initial PCR amplification of 16s RNA genes was performed using Taq DNA polymerase (New England Biolabs, Ipswich, United States) with the following conditions: an initial denaturation at 95 °C for 30 s, followed by 30 cycles consisting of denaturation at 95 °C for 10 s, annealing at 55 °C for 30 s and extension at 72 °C for 30 s, and a final extension step at 72 °C for 5 min. A PCR clean-up was carried out using Agencourt Ampure XP beads (Beckman Coulter, Brea, United States) prior to performing index PCR using Nextera XT index kit (Illumina, San Diego, United States). The amplicons were sequenced using the $2 \times 250$ paired-end protocol (MiSeq Reagent Kit v2 for 500 cycles) on an Illumina MiSeq instrument.

## Data analysis

The MiSeq-generated sequence data were trimmed via Sickle (http://github.com/najoshi/sickle) using the following parameters: -q 20 –l 200. Merging of paired-end sequences, filtering of chimeric sequences, clustering of 16S rRNA sequences into Operational Taxonomic Units (OTUs) and taxonomic classification of OTU were conducted using Micca version 1.6.2 (*Albanese et al., 2015*). Briefly, micca mergepairs was run to merge paired-end sequences with a minimum overlap length of 25 bp and maximum allowed mismatches of 5. The output sequences were then subjected to micca filter to retain sequences that have a minimum length of 400 bp and an expected error rate of not greater than 0.5%. Subsequently micca otu was executed to allow filtering of chimeric sequences and *de novo* clustering of sequences into OTUs with a 97% identity threshold.

Taxonomic assignment of the *de novo*-clustered OTUs was performed using the Bayesian LCA-based taxonomic classification method with a 1e−100 cut-off *e*-value and 100 bootstrap replications, against the NCBI 16S microbial database (*Coordinators, 2018*; *Gao et al., 2017*). Taxonomic assignment at each level was accepted only with a minimum confidence score of 80. Multiple sequence alignment of the OTU representative sequences was performed using PASTA (*Mirarab et al., 2015*). A phylogenetic tree was constructed using FastTree under the GTR+CAT model (*Price, Dehal & Arkin, 2010*).

The rarefaction depth values were set at 5,249 and 3,305, respectively, for the overnight and daytime samples, prior to further analysis using QIIME (version 1.9.1) to compare

**Table 1  Demographic characteristics of volunteers recruited in this study.**

| Participant characteristics | UBT-positive (N = 10) | UBT-negative (N = 14) |
|---|---|---|
| Age | | |
| 20–29 | 4 | 7 |
| 30–39 | 4 | 6 |
| 40 and above | 2 | 1 |
| Gender | | |
| Male | 5 | 9 |
| Female | 5 | 5 |

alpha and beta diversities between UBT-positive and UBT-negative individuals (*Caporaso et al., 2010*). Alpha diversity was evaluated based on the following metrics: observed species, Shannon index and Chao1 index. Non-parametric two-sample $t$-test with Bonferroni correction was used to compare the alpha diversity metrics between groups. Principle Coordinates Analysis (PCoA) using weighted UniFrac distance metric was performed to visualize separation of samples. Non-parametric statistical analysis of the distance metric was performed using ANOSIM with 1000 permutations.

## Statistical analysis

LEfSe was employed to identify distinguishing bacterial phyla and genera between the *H. pylori-* positive and negative cohorts (*Segata et al., 2011*).

## Accession numbers

All sequencing data generated in this study are available in the Sequence Read Archives (SRA) database under the accession numbers listed in Table S1.

## RESULTS

Among the 24 participants who underwent the Carbon-14 Urea Breath Test, 10 and 14 were determined as *H. pylori* positive and negative, respectively. The demographic characteristics of the participants are summarised in Table 1.

### *16S rRNA* gene sequencing

A total of 5,561,542 and 8,258,347 reads were generated for the overnight and daytime samples, respectively. Following quality trimming, merging and filtering of paired-end sequences, 628,351 with an average read length of 461.5 ± 7.5 bp were retained for the overnight samples. For the daytime samples, 906,743 reads with an average length of 461.8 ± 7.3 bp were obtained. These reads were further clustered into 146 and 137 OTUs for the overnight and daytime datasets, respectively, after the removal of chimeric sequences.

### Characterization of the oral microbiota

The bacterial communities in the UBT-negative and UBT-positive cohorts of both periods were analysed at phylum and genus levels (Tables S2 and S3, respectively). During the day, Firmicutes, Proteobacteria and Actinobacteria constituted the top three most abundant

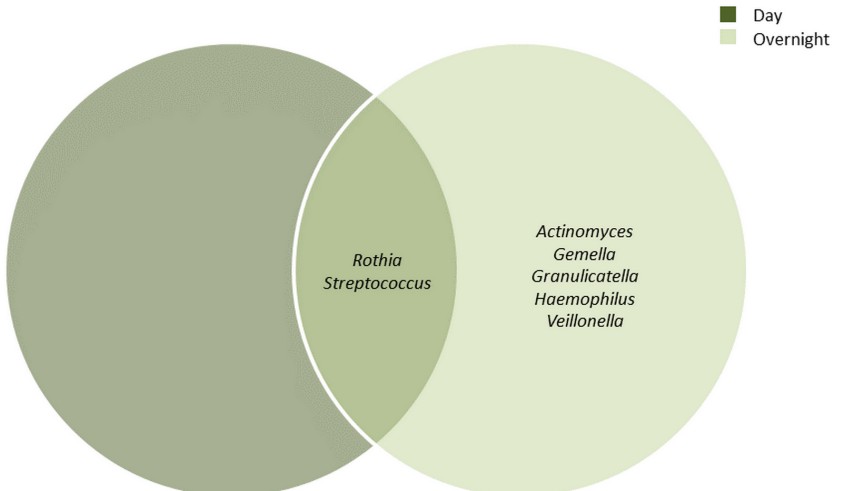

**Figure 1** **Core bacterial genera identified among the daytime and overnight samples.** A minimum threshold of 0.9 was employed for the identification of core genera among the daytime and overnight samples.

phyla, accounting for 43.9%, 34.5% and 17.7% of the oral microbiota, respectively. In the overnight samples, the top three most abundant phyla were Firmicutes, Proteobacteria and Bacteroidetes, with relative abundances of 51.9%, 29.2% and 10.8%, respectively. At the genus level and by using a minimum frequency threshold of 90%, *Rothia* and *Streptococcus* were identified in most samples of both time periods, with daytime relative abundances of 14.2% and 30.4%, and 5% and 42.6% in overnight, respectively (Fig. 1). In addition to the two genera mentioned above, *Actinomyces*, *Gemella*, *Granulicatella*, *Haemophilus* and *Veillonella* were shared between most overnight samples, together representing 16.2% of the oral microbiota. Notably, we were unable to detect any *Helicobacter* sequences from our samples in this study.

## Diversity and LEfSe analyses

To estimate the alpha diversity of the oral microbial community, we employed observed species, Shannon diversity and Chao1 richness estimator indexes. As depicted in Fig. 2, the rarefaction curves of Shannon diversity index for all daytime and overnight plateaued, indicating that there is sufficient coverage of microbial diversity per sample. Further comparisons between the UBT-positive and UBT-negative cohorts by the Shannon diversity index demonstrated that during the day, the former exhibits significantly reduced oral microbial diversity compared with that of the latter ($p$-value = 0.03) (Fig. 3). At night, however, no significant difference in microbial diversity could be observed.

Principal Coordinates Analysis (PCoA) based on weighted UniFrac index was performed at the OTU level to assess if there is any grouping of the samples by sharing similar bacterial communities. No obvious separation was observed in the overnight samples (Fig. 4). In the daytime samples, however, we observed a good clustering of the bacterial communities in the UBT-negative cohort (Fig. 5). The UBT-positive samples displayed an irregular

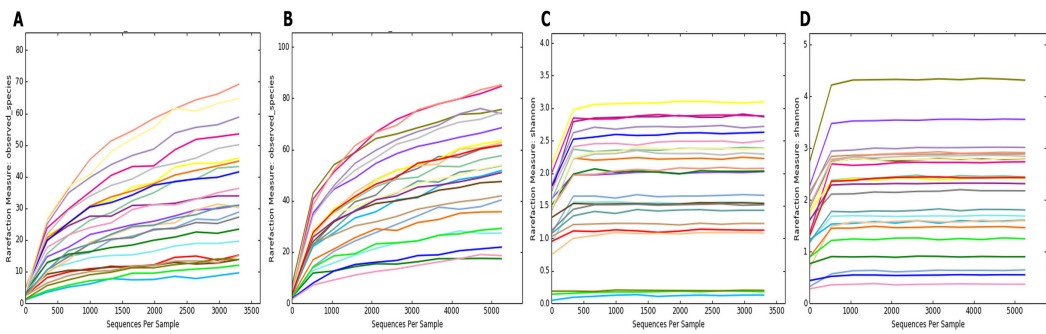

**Figure 2   Rarefaction curve plots of the daytime and overnight samples.** (A) Rarefaction curve of daytime samples based on the numbers of observed species. (B) Rarefaction curve of overnight samples based on the numbers of observed species. (C) Rarefaction curve of daytime samples based on Shannon diversity index. (D) Rarefaction curve of overnight samples based on Shannon diversity index.

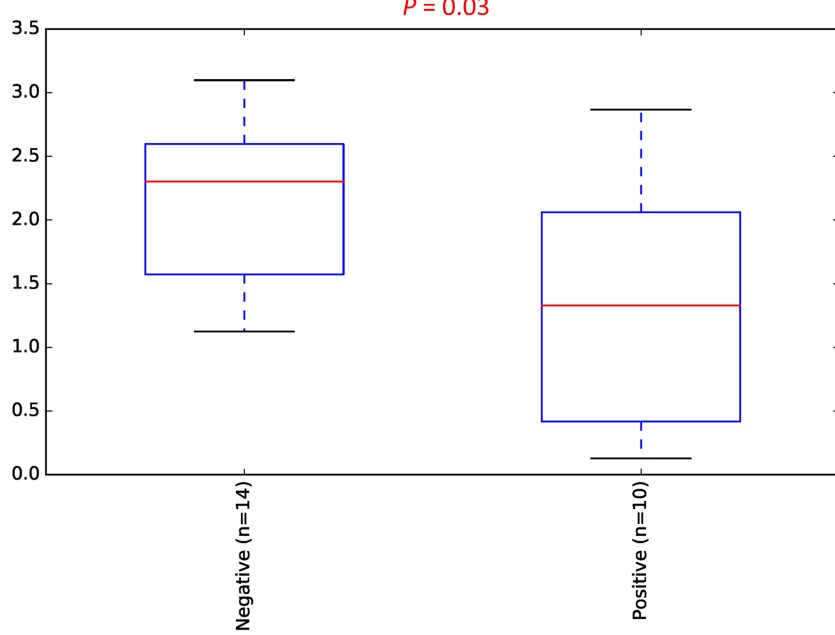

**Figure 3   Daytime comparison of microbial diversity between the UBT-negative and UBT-positive cohorts.** Boxplot of Shannon index depicts significant differences in daytime bacterial diversity between the UBT-negative and UBT-positive cohorts.

distribution pattern, suggesting that *H. pylori* infection disturbs the oral microbiome during the day. An analysis of similarities (ANOSIM) non-parametric statistical test based on weighted UniFrac distance further revealed that there is a significant difference in the bacterial community composition between the cohorts during the day ($p$-value $= 0.003$, $R = 0.343$).

To identify which taxa could be significantly affected within the cohorts in the daytime, we employed the linear discriminant analysis effect size (LEfSe) method. At the

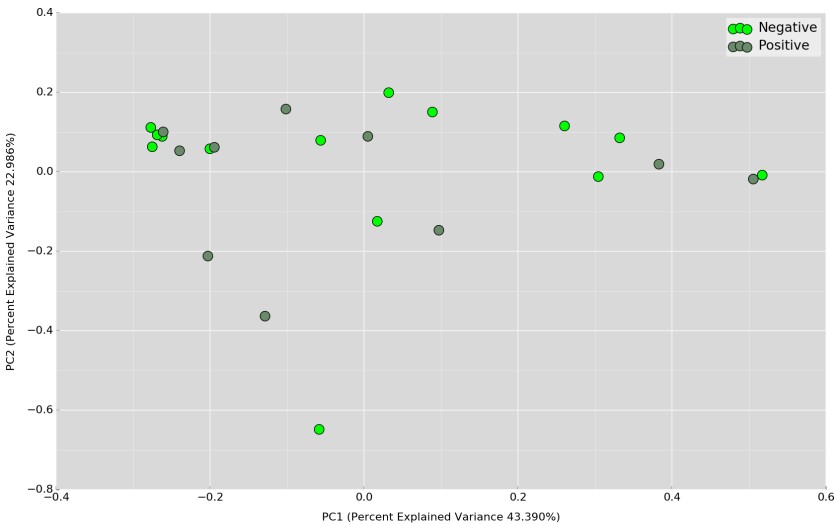

**Figure 4** PCoA plot based on weighted UniFrac distance metric comparing overnight OTU abundances between UBT-positive and UBT-negative cohorts.

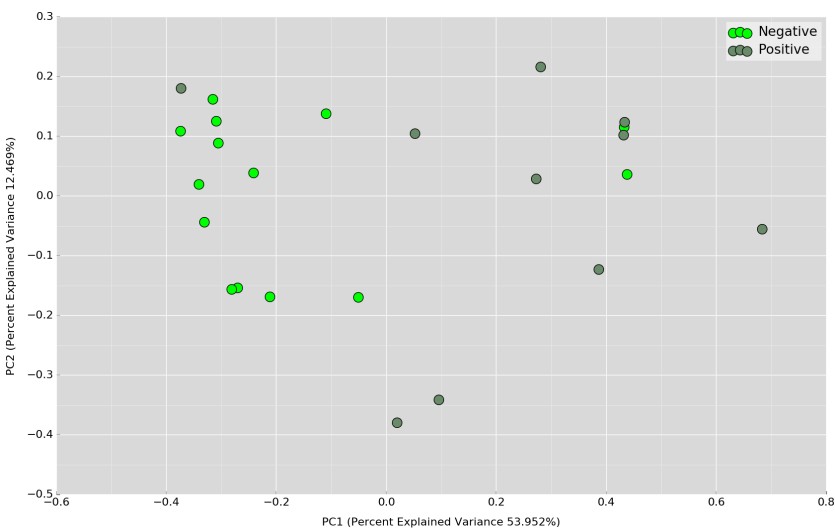

**Figure 5** PCoA plot based on weighted UniFrac distance metric comparing daytime OTU abundances between UBT-positive and UBT-negative cohorts.

phylum level, Proteobacteria were significantly enriched in UBT-positive samples whilst Fusobacteria and Firmicutes were significantly reduced (Fig. 6A). At the genus level, in UBT-positive samples, significantly higher abundances were detected in both *Pseudomonas* and *Roseomonas*, while *Fusobacterium*, *Solobacterium*, *Haemophilus* and *Streptococcus* were significantly decreased (Fig. 6B). We attempted the analysis on the overnight samples, but identified no distinguishing taxa, in agreement with the PCoA outcome which also showed no obvious separation between cohorts.

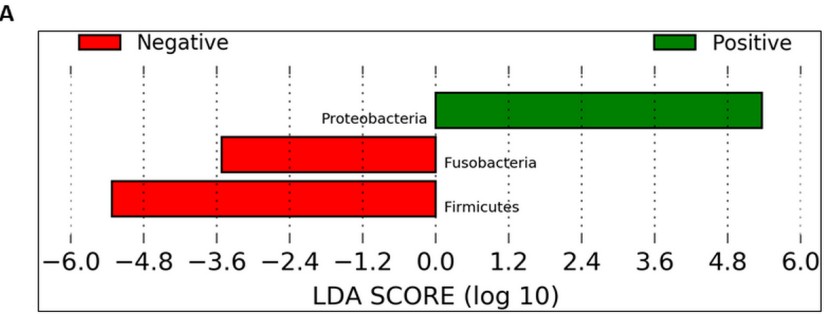

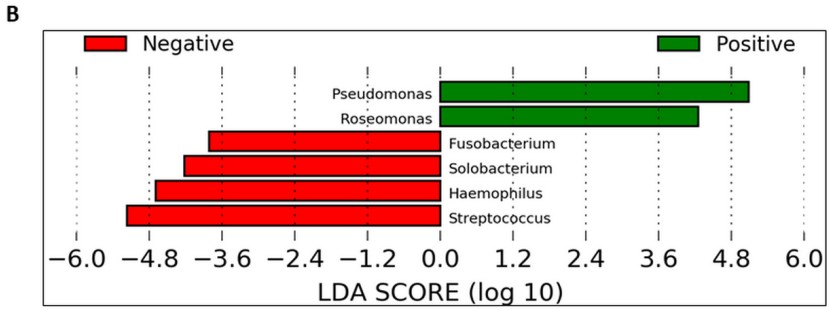

**Figure 6 Identification of bacterial phyla and genera with significant differences between the UBT-negative and UBT-positive daytime samples using LEfSe analysis.** (A) Significantly differentially abundant bacterial phyla identified by LEfSe analysis. (B) Significantly differentially abundant bacterial genera identified by LEfSe analysis.

## DISCUSSION

In this study, the effect of *H. pylori* infection on the oral microbiota was explored. In our sequencing data, no trace of *Helicobacter 16S rRNA* gene sequence could be detected at either time point among the 10 *H. pylori*-positive volunteers recruited in this study. We also attempted PCR detection using *H. pylori*-specific primers targeting *16S rRNA*, *ureI* and *vacA* genes. However, these yielded no positive outcomes (data not shown). In a recent meta-study, it was demonstrated that the presence of *H. pylori* in saliva detected via either PCR or culture methods was less than that from dental plaque (*Anand, Kamath & Anil, 2014*). Hence the absence of *Helicobacter* sequence in our study could be attributed to our oral swab collection method that did not include any dental plaque material, or it may be that *H. pylori* was genuinely absent in the oral cavity of these UBT-positive individuals on the days covered by the samples.

The core bacterial genera identified in our study were in agreement with the findings elsewhere reported (*Bik et al., 2010*; *Seoudi et al., 2015*). *Actinomyces*, *Gemella* and *Veillonella* were more prevalent in the overnight samples, consistent with their anaerobic characteristic (*Delwiche, Pestka & Tortorello, 1985*; *Valour et al., 2014*; *Vasishtha, Isenberg & Sood, 1996*). The remaining genera including *Rothia*, *Streptococcus*, *Granulicatella* and *Haemophilus* have been shown to contain most, if not all, species that are facultative anaerobes (*Collins & Lawson, 2000*; *Kuklinska & Kilian, 1984*; *Tunney et al., 2008*).

Despite the fact that the presence of *Helicobacter* genus could not be detected in any of our samples, significant differences in both alpha and beta diversities were observed in *H. pylori*-positive individuals relative to the negative comparison group during the day. It has been shown that *H. pylori* infection is associated with the systemic changes of different metabolic and immunological factors, so it is possible that any of these changes would perturb the stability of the oral microbial community (*Buzas, 2014*). It is also interesting to observe that such changes targeted only the daytime oral microbiota but not those at night. There has been abundant evidence that *H. pylori* is capable of colonising the oral cavity and since *H. pylori* is an obligate aerobe, a possible explanation would be that during the day, the proliferation of *H. pylori* changes the nutrient availability and possibly pH within the oral cavity, in addition to the secretion of *H. pylori* proteins and metabolites (*Anand, Kamath & Anil, 2014*). These possible events, together, may affect the growth of certain bacterial species and thus altering the structure of the daytime oral microbiota.

Further LEfSe analysis at the genus level revealed six genera that were different between both UBT-positive and UBT-negative cohorts in the daytime. Among the four genera that exhibited remarkably lower abundances in the UBT-positive samples, *Fusobacterium* and *Solobacterium* have been implicated in periodontal disease (*Schirrmeister et al., 2009*; *Signat et al., 2011*). *Streptococcus*, on the hand, contains oral members that are opportunistic pathogens commonly associated with the development of cardiovascular diseases (*Adam et al., 2015*; *Nakano et al., 2009*). It is noteworthy that both *Fusobacterium* and *Solobacterium* are often associated, and that *Fusobacterium* and *Streptococcus* could form aggregates with each other, reflecting a potentially synergistic and/or symbiotic relationship between them that may enhance bacterial survival fitness for long term colonisation of the human oral cavity (*Edwards, Grossman & Rudney, 2006*; *Kaplan et al., 2009*; *Schirrmeister et al., 2009*). Meanwhile, albeit statistically significant, the clinical importance of *Solobacterium* should be carefully examined owing to its low sequence counts among the UBT-negative samples. Of the significantly diminished *Haemophilus* bacterial population in our UBT-positive samples, four species were identified including *H. haemolyticus*, *H. paraphrohaemolyticus*, *H. pittmaniae* and *H. parainfluenzae*, which are common bacterial flora of the oral cavity (*Kuklinska & Kilian, 1984*). While *Haemophilus* spp. are generally regarded as harmless to human host, *H. parainfleunzae* has been reported as an opportunistic pathogen for causing respiratory tract infections and endocarditis (*Darras-Joly et al., 1997*; *Rhind et al., 1985*). The apparent inhibitory effect of *H. pylori* infection on the growth on these opportunistic pathogens in our study, but only in the daytime, is intriguing. It is uncertain whether such inhibition would reduce the risk of developing diseases associated with these bacteria.

In this study, *Pseudomonas* and *Roseomonas* were significantly enriched in the UBT-positive samples. While members of the genus *Pseudomonas* rarely cause disease in healthy individuals and only *P. aeruginosa* has been widely reported on its role in respiratory tract infection in immunocompromised patients, the latter was not detected in our UBT-positive samples (*Rivas Caldas & Boisrame, 2015*). It is noteworthy that no *Pseudomonas* sequence was present in any of the UBT-negative samples. Although the *Pseudomonas* species identified in this study had no identified disease correlations, their presence in the oral cavity of *H. pylori* infected individuals, and thus the potential impact on one's general

health, should be carefully assessed. Similar to *Pseudomonas*, human infections caused by *Roseomonas* species are uncommon (*Wang et al., 2012*). The detection of *Roseomonas* only among the UBT-positives, but not in any UBT-negatives, during the day is again interesting. Nevertheless, like *Solobacterium*, its clinical relevance should be interpreted cautiously due to its very low abundance in *H. pylori* infected individuals.

## CONCLUSIONS

Our study reported the effects of *H. pylori* infection on the human oral microbiome, particularly during the day. The abundance levels of several bacterial genera including opportunistic pathogens were significantly affected by *H. pylori* infection. To better understand and validate the effects of *H. pylori* infection on the human oral microbiota, another study entailing a greater sample size would be necessary. Also, analysing the oral microbiota with the same person after eradication would help to resolve whether *H. pylori* infection directly affects the oral microbiome. Further studies may be required to elucidate the impacts of these bacterial community changes on the general health of individuals with *H. pylori* infection.

## ACKNOWLEDGEMENTS

We would like to thank Tri-Med Distributors Pty Ltd for providing the UBT screening service.

### Funding

This study was funded by ShenZhen's Sanming Project (Grant No: SZSM201510050) and the Vice Chancellor of the University of Western Australia. The funders had no role in study design, data collection and analysis, decision to publish, or preparation of the manuscript.

### Grant Disclosures

The following grant information was disclosed by the authors:
ShenZhen's Sanming Project: SZSM201510050.
Vice Chancellor of the University of Western Australia.

### Competing Interests

The authors declare that there are no competing interests.

### Author Contributions

- Eng-Guan Chua analyzed the data, prepared figures and/or tables, authored or reviewed drafts of the paper, approved the final draft.
- Ju-Yee Chong and Binit Lamichhane performed the experiments, approved the final draft.
- K. Mary Webberley authored or reviewed drafts of the paper, approved the final draft, ethics application.

- Barry J. Marshall contributed reagents/materials/analysis tools, approved the final draft.
- Michael J. Wise conceived and designed the experiments, analyzed the data, authored or reviewed drafts of the paper, approved the final draft.
- Chin-Yen Tay conceived and designed the experiments, authored or reviewed drafts of the paper, approved the final draft, ethics application.

## Human Ethics

The following information was supplied relating to ethical approvals (i.e., approving body and any reference numbers):

This study was approved by the Human Ethics of the University of Western Australia (RA/4/1/7953). Written consent was obtained from all individuals participated in this study.

## Data Availability

All sequencing data generated in this study are available in the Sequence Read Archives (SRA) database. Their accession numbers are available in Table S1.

## Supplemental Information

Supplemental information for this article can be found online at http://dx.doi.org/10.7717/peerj.6336#supplemental-information.

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
