# Peer review of "Gastric Helicobacter pylori infection perturbs human oral microbiota"

_PeerJ, doi:10.7717/peerj.6336_

## Round 0.1 · original submission · Major Revisions

Dear Dr. Chua and colleagues:

Thanks for submitting your manuscript to PeerJ. I have now received two independent reviews of your work, and as you will see, the reviewers raised some concerns about the research. Despite this, these reviewers are optimistic about your work and the potential impact it will have on the Helicobacter and oral pathogens research communities. Thus, I encourage you to revise your manuscript accordingly, taking into account all of the concerns raised by both reviewers.

While the concerns of the reviewers are relatively minor, this is a major revision to ensure that the original reviewers have a chance to evaluate your responses to their concerns.

I look forward to seeing your revision, and thanks again for submitting your work to PeerJ.

Good luck with your revision,

-joe

Reviewer 1 ·

Basic reporting

no comment

Experimental design

no comment

Validity of the findings

no comment

Additional comments

As understood from the paper, the authors have tried to understand the effect of H. pylori infection on the composition of the oral microbiota. I have the following comments on the manuscript:

The epidemiological data provided lacks details such as
a) The study duration
b) The timeframe for recruitment of the volunteers i.e were they recruited together or was there a difference in time for sample collection?
c) Whether the volunteers suffered from any other comorbidities such as gastritis or previously undertook medications in the recent past.
d) Since H. pylori is associated with ethnicity it would be important to know the same for study volunteers.

It is also not clear the why the authors reasoned that H. pylori infection might have an influence on the microbiota? Or should be reworded to mean that the aim of the study was to understand whether the oral microbiota differs between UBT positive vs UBT negative individuals.
It appears from the observations made that there could be differences in the microbiota of UBT positive versus UBT negative individuals that might be an indirect effect or due to comorbidities which is difficult to resolve given the size of the study.
The hypothesis laid down in the introduction was that H. pylori is present in the oral cavity, but in the samples studied the authors did not identify H. pylori in oral swabs. In addition, since the difference observed is only limited to the daytime samples only points to the fact that it's not a consistent phenomenon. Therefore, it would not be fair to associate H. pylori positivity to the observed changes and would need a carefully designed study with a greater number of samples.
The manuscript needs a careful rewording the project the findings supported by the observations made.

·

Basic reporting

No concerns

Experimental design

No comment

Validity of the findings

Analyses should be done at the Family and/or Order level, since patient-to-patient variability at the genus level is so large that most differences aren’t meaningful. It may be possible to find more meaningful differences at higher taxonomic levels.

Additional comments

Major comments:
1. Lines 154 and 155: The sentence states that 14 were H. pylori positive and 10 were negative. This should be reversed, since 10 were positive.
2. The demographic table doesn’t add up. Under gender, it says that 9 were male and 2 were female. This should add up to 14.

Minor comments:
1. The rate of infection is much higher than expected for Australia. Were many of the subjects foreign nationals from countries with higher infection rates?
2. Roseomonas was only detected in 3 patients and only at 1 or 2 sequences per patient. This may pass the statistical test, but its relevance seems questionable at best. The situation with Solobacterium isn’t much better.
3. Lines 235-236: The authors state that it is difficult to explain why microbiota changes are seen during the day, but not at night. There is abundant evidence that H. pylori colonizes the mouth to some degree. Since H. pylori is an obligate aerobe, more H. pylori growth would occur during the day, thus changing nutrient availability, H. pylori proteins and metabolites, and possibly pH within the oral cavity.
4. It would be useful to determine the prevalence of Streptococcus mutans. Reductions in S. mutans could prevent dental caries. Also, urease production by H. pylori could reduce oral acidity, helping to protect teeth.

---

## Round 0.2 · Minor Revisions

Dear Dr. Chua and colleagues:

Thanks for re-submitting your manuscript to PeerJ. The two reviewers are happy with your revision but have raised two important concerns. I believe that addressing these concerns will make your manuscript ready for publication In PeerJ.

Good luck with your minor revision,

-joe

Reviewer 1 ·

Basic reporting

no comment

Experimental design

no comment

Validity of the findings

no comment

Additional comments

I agree with the authors reply on the previous comments. I have only one suggestion to include a paragraph detailing the limitations of the study clearly especially the small sample size.

·

Basic reporting

No comment

Experimental design

No comment

Validity of the findings

No comment

Additional comments

The authors have responded appropriately to reviewer comments.

Minor comments
I do not understand why the authors have changed the color schemes in figures 1, 4, and 5. It is much more difficult to differentiate between light green and dark green than between blue and red. The schemes should be changed to improve contrast.

---

## Round 0.3 · accepted · Accept

Dear Dr. Chua and colleagues:

Thanks for re-submitting your manuscript to PeerJ, and for addressing the concerns raised by the reviewers. I now believe that your manuscript is suitable for publication. Congratulations! I look forward to seeing this work in print, and I anticipate it being an important resource for the communities studying Helicobacter pylori infection and human oral microbiota. Thanks again for choosing PeerJ to publish such important work.

-joe

#